# Effects of digital cognitive behavioral therapy for depression on suicidal thoughts and behavior: Protocol for a systematic review and meta-analysis of individual participant data

**Lasse Bosse Sander**[1,2]*, **Marie Beisemann**[3], **Eirini Karyotaki**[4], **Wouter van Ballegooijen**[4,5], **Pim Cuijpers**[4], **Tobias Teismann**[6], **Philipp Doebler**[3], **Matthias Domhardt**[7], **Harald Baumeister**[7], **Rebekka Büscher**[1]

1 Medical Psychology and Medical Sociology, Faculty of Medicine, University of Freiburg, Freiburg, Germany, 2 Black Dog Institute, University of New South Wales, Sydney, New South Wales, Australia, 3 Department of Statistics, TU Dortmund University, Dortmund, Germany, 4 Department of Clinical, Neuro and Developmental Psychology, Amsterdam Public Health Research Institute, Vrije Universiteit Amsterdam, Amsterdam, The Netherlands, 5 Department of Psychiatry, Amsterdam UMC, Amsterdam, The Netherlands, 6 Mental Health Research and Treatment Center, Faculty of Psychology, Ruhr-Universität Bochum, Bochum, Germany, 7 Institute of Psychology and Education, Department of Clinical Psychology and Psychotherapy, Ulm University, Ulm, Germany

* Lasse.Sander@mps.uni-freiburg.de

**Data Availability Statement:** No datasets were generated or analyzed during the current study.

## Abstract

### Introduction

Digital cognitive behavioral therapy (i-CBT) interventions for the treatment of depression have been extensively studied and shown to be effective in the reduction of depressive symptoms. However, little is known about their effects on suicidal thoughts and behaviors (STB). Information on the impact of digital interventions on STB are essential for patients' safety because most digital interventions are self-help interventions without direct support options in case of a suicidal crisis. Therefore, we aim to conduct a meta-analysis of individual participant data (IPDMA) to investigate the effects of i-CBT interventions for depression on STB and to explore potential effect moderators.

### Methods

Data will be retrieved from an established and annually updated IPD database of randomized controlled trials investigating the effectiveness of i-CBT interventions for depression in adults and adolescents. We will conduct a one-stage and a two-stage IPDMA on the effects of these interventions on STB. All types of control conditions are eligible. STB can be measured using specific scales (e.g., Beck scale suicide, BSS) or single items from depression scales (e.g., item 9 of the PHQ-9) or standardized clinical interviews. Multilevel linear regression will be used for specific scales, and multilevel logistic regression will be used for treatment response or deterioration, operationalized as a change in score by at least one quartile from baseline. Exploratory moderator analyses will be conducted at participant,

The data availability of the subsequent study will be handled when the study is complete according to open access principles and legal options/individual participants rights.

**Funding:** LBS received funding for the study described in this protocol from the German Research Foundation (project "Effects of internet interventions for depression on suicide ideation - a meta-analysis of individual participant data (IPDMA), grant No. SA 3767/2-1). URL: http://www.dfg.de The article processing charge was funded by the German Research Foundation (DFG) and the University of Freiburg in the funding program Open Access Publishing. The funders had no role in the study design, the decision to publish, or in the preparation of the manuscript. The views expressed are those of the author(s) and not necessarily those of the funder.

**Competing interests:** The authors have declared that no competing interests exist.

study, and intervention level. Two independent reviewers will assess the risk of bias using the Cochrane Risk of Bias Tool 2.

## Conclusion

This IPDMA will harness the available data to assess the effects (response and deterioration) of i-CBT interventions for depression interventions on STB. Information about changes in STB is essential to estimate patients' safety when engaging in digital treatment formats.

## Trial registration

We will pre-register this study with the open science framework after article acceptance to ensure consistency between online registration and the published trial protocol.

## Introduction

Depression is one the largest causes of disability, estimated to affect about 280 million people worldwide [1]. A well-established evidence base indicates that various forms of psychotherapy are effective in treating depression [2–6]. Many patients prefer psychotherapy as the first line of treatment [7, 8], which outperforms medication in terms of long-term effectiveness and side effects [9–11]. However, the scalability of psychotherapy is limited due to several barriers, including high treatment costs, a shortage of trained clinicians, and stigma [12–14]. Therefore, the developing of new treatment approaches and scalable formats is an urgent priority.

To meet this challenge, extensive research efforts have been conducted in digital behavioral interventions over the past two decades [15, 16]. Digital interventions are often self-help interventions that include psychotherapeutic content and exercises in the form of text, audio, video, and interactive elements in an app or on a website. Additional homework assignments and automated reminders or feedback are thought to promote behavior change and the integration of learned exercises into everyday life [17, 18]. Digital interventions can be delivered as fully self-help interventions or with human guidance or support. Human guidance can range from brief feedback on completed modules, clarifying of content, and motivating participants (also referred to as guided self-help) to full parallel therapy sessions in face-to-face contact in the form of blended care.

Although digital mental health is a relatively young field of research, there is already a well-established evidence base documenting the effectiveness of digital interventions in the prevention and treatment of depression [2, 19]. The effect sizes of guided digital interventions are comparable to those of face-to-face therapies [2, 20].

In addition to these positive results, there are other advantages of digital interventions, including the ability to use interventions at a time that suits the user, scalability, and anonymity. Given these benefits, there have been considerable efforts in many countries to implement digital interventions in healthcare settings [21, 22]. However, a much-debated question in both research and practice is whether digital interventions are also suitable for people with depression who have suicidal thoughts and behavior (STB). STB are common in people suffering from depression and a key diagnostic criterion in both DSM-5 and ICD-11 [23–25].

The use of digital interventions for people with depression and STB may raise some concerns, such as limited clinical impression of the patient, limited access to patients in crisis, and potential delays in accessing established face-to-face treatments [26, 27]. Furthermore, a strong therapeutic relationship is believed to be a crucial component in the treatment of people with

STB [28–30], which may be challenging to achieve to the same extent as in traditional face-to-face treatments when using digital formats.

Meta-analyses have shown that i-CBT targeting suicidality ('direct interventions') can effectively reduce suicidal ideation, with modest effect sizes [31–33]. Conversely, Torok and colleagues concluded in their meta-analysis of aggregated data that 'indirect interventions' (i.e., interventions targeting depression and only indirectly address suicidality) did not significantly reduce suicidal ideation [31]. However, this meta-analysis has limitations in that it does not represent digital interventions for depression, as it only includes six out of more than eighty conducted trials [2]. Most trials were excluded because they did not use a specific diagnostic tool to assess suicidal ideation [31]. Additionally, in these six trials, elevated symptoms of suicidality were not an inclusion criterion, which likely resulted in a sample where many participants did not experience suicidal ideation at baseline, thereby diminishing the potential to identify an effect on this outcome.

Meta-analyses of aggregated data cannot investigate effects at the participant level or examine participant-level moderators, as they are limited to reported group differences [34, 35].

Against this backdrop, we plan to conduct an individual patient data meta-analysis (IPDMA) to investigate treatment effects of i-CBT interventions for depression on STB.

IPDMAs merge the original research data from primary trials, enabling more detailed and robust analyses of combined primary participant data on an item level [36, 37].

Further, IPDMA offer several advantages over the analysis of aggregated data, such as the use of unreported data, integrity checks of trial IPD, standardization of outcome measures and participant characteristics, and risk of bias assessments based on trial IPD. Moreover, IPDMAs outperform the meta-analyses of aggregated data in terms of the range of outcome measures, the possibility of validity checks of analysis assumptions, the comparability by using consistent methods for the analysis, and the option to apply non-standard effect measures [36].

The proposed IPDMA will leverage an existing IPD pool of currently 62 studies (14,088 participants) on i-CBT interventions for depression [20, 38, 39]. Currently, this dataset is undergoing a major update, and the number of included studies and participants is expected to alter. We will investigate treatment effects of i-CBT depression interventions on STB, while also exploring potential moderators of this effect.

## Materials and methods

This study has been pre-registered with OSF. (Note: We will pre-register this study with the open science framework after article acceptance to ensure consistency between online registration and the published trial protocol). This study will be reported in accordance with the PRISMA protocols statement [40] and the PRISMA IPD guideline [41]. Any potential amendments to the protocol will be reported in the final report.

### Eligibility criteria

Please refer to Table 1 for an overview of the PICOS (participants, interventions, comparators, outcomes, study design) elements of the study inclusion criteria.

**Population.** This review will include studies that focus on (a) adults or adolescents with (b) depressive symptoms (including clinical and subthreshold symptoms) as determined by either a cut-off score on self-report outcome measures or a diagnostic interview.

**Interventions.** Studies will be eligible if the experimental condition involves cognitive behavioral therapy for depression delivered via the web or a mobile application. Interventions

**Table 1. PICOS (participants, interventions, comparators, outcomes, study design) elements of the study inclusion criteria.**

**Participants**
- Adults or adolescents with depressive symptoms

**Interventions**
- Digital (web- or mobile-based) cognitive behavioral therapy interventions for depression with or without human support (guidance).

**Comparators**
- Usual care
- Attention control/psychological placebo
- Active control group
- No intervention
- Waitlist

**Outcomes**
- Depressive symptoms assessed by diagnostic interviews, self-reported and clinician -rated scales
- Suicidal thoughts and behaviors
- Self-report scales
- Diagnostic interviews
- Single items of depression scales (e.g., item 9 of the, PHQ-9)

**Study design**
- Randomized controlled trials published in peer-reviewed journals.

may involve human support through the internet, either synchronous or asynchronous. We will exclude interventions that specifically target suicidal ideation; however, interventions that exclusively apply safety procedures or provide emergency contacts will not be excluded. Blended care, i.e., internet-based interventions combined with face-to-face treatment, will be excluded.

**Comparators.** Trials with the following control conditions will be eligible: usual care, attention control/psychological placebo, active control group, no intervention or waitlist.

**Outcomes.** Studies will be included if they report a quantitative measure of depressive symptoms. Depressive symptoms may be assessed by diagnostic interviews (e.g., Structured Clinical Interview, SCID [42]), clinician-rated scales (e.g., Hamilton Depression Rating Scale, HAMD [43]), or self-report scales (e.g., Beck Depression Inventory, BDI [44]) for depression. Additionally, a quantitative measure of STB at baseline and post-intervention must be available in the published article and/or the IPD. STB may be measured via self-report, diagnostic interviews, or single items of depression scales (e.g., item 9 of the Patient Health Questionnaire 9, PHQ-9 [45]).

**Studies.** Only randomized controlled trials (RCTs) published in a peer-reviewed journal will be eligible for inclusion. There will be no restrictions on language or publication date.

## Study identification and data extraction

We will use an existing database of RCTs investigating psychological interventions for depression, which has been described in detail elsewhere [20, 38, 39]. The database is updated annually. IPD is sought irrespective of whether the published article report STB assessments. We will extract deidentified IPD from all randomized participants, as well as relevant data items from published articles, including study identification items, study procedures, and aggregated

clinical and sociodemographic participant data. This will include data on the theoretical foundation of the intervention, delivery mode and content, amount of human support in remote treatments, and qualification of healthcare providers.

The primary outcome of this study will be the effects of the interventions on STB assessed by specific scales (e.g., Beck Scale for Suicide Ideation [46]), single-item analyses from depression scales, or standardized interviews (e.g., suicidality item of the HAMD). We will use both specific scales and single items in separate analyses where available. If multiple validated measures of STB are available, we will prefer self-reports over clinician ratings, as individuals are more likely to report existing STB in self-report [47, 48]. If multiple measures of the same hierarchy level exist within a study, we will either extract the measure most frequently used across eligible studies or randomly select if not evident.

## Risk of bias

Risk of bias will be assessed using the RoB 2.0 (revised Cochrane Risk of Bias Tool for randomized controlled trials) [49]. Five domains will be evaluated for potential bias: randomization process, deviations from planned interventions, missing outcome data, measurement of the outcome, and selection of the reported result. We will not assess items related to blinding of outcome assessors (signaling questions 4.3 to 4.5), because the tool defines participants as outcome assessors for outcomes that cannot be measured without incorporating the participant's perspective, and blinding is rarely possible in psychotherapy research. This will prevent ceiling effects (i.e., overall high risk of bias for all studies) and allow for distinguishing between studies of high and low quality in this research field. Bias ratings will be based on information from published trials and IPD, with the latter potentially altering the judgment of bias potential, such as selecting an appropriate and pre-registered analysis strategy in IPDMA, leading to a lower risk of bias in the selection of the reported result compared to the original trial. We will also check for potential range restrictions and high or low variances in the primary outcome, particularly relevant for single item measures.

Moreover, we will assess availability bias by examining whether the studies included in this IPDMA with available IPD and STB measures are representative of all relevant trials. First, we will conduct a random-effects meta-analysis of aggregated data with depressive symptoms as the outcome using the classic inverse-variance approach, performing a subgroup analysis to compare studies included in this IPDMA with all other relevant trials that did not provide IPD with STB measure. The between-groups effect will be calculated using change scores from baseline. Second, we will display study characteristics for included trials and other relevant trials, such as year of publication, treatment format, number of sessions, age, gender, recruitment strategy, control condition, to identify any group differences.

## Quality of evidence

Two independent reviewers will use the Grading of Recommendations Assessment, Development and Evaluation (GRADE) to assess the quality of evidence for the treatment effect on STB. The following domains will be evaluated: risk of bias, inconsistency, indirectness of evidence, imprecision, and publication bias. Any discrepancies will be resolved through discussion with a third researcher.

## Statistical analyses

Although STB is the primary outcome of this study, it is usually not the primary outcome of depression intervention studies, which poses several methodological challenges. First, only a subset of eligible studies will have a validated measure of STB, and most studies will likely have

single-item measures of STB as part of a depression measure. Since single items have only a few levels, they cannot be interpreted as continuous outcomes, which limits the range of potential analyses strategies. Second, individuals with STB and/or a history of suicide attempts are often excluded from depression trials, leading to a floor effect that limits the ability to demonstrate potential reductions in STB [27].

In our analyses, we will utilize all available data to estimate the effect on STB. We will analyze STB using three indices: (1) treatment response and (2) symptom deterioration for single-item measures, and (3) a continuous score for the studies that provide validated questionnaires for STB. Severity of STB will be operationalized as a continuous post-intervention score. We will define treatment response versus deterioration as an improvement or deterioration by at least one quartile from baseline on the relevant item. We chose not to use a commonly used 50% change criterion because this would equate a minor improvement in the low scale range with a larger improvement in the high scale range. Furthermore, the potential to reach a change of one quartile across different single-item measures may differ. Therefore, we will conduct sensitivity analyses for each measuring instrument (see sensitivity analyses below). All analyses will be performed using R.

With regard to missing data handling, we will perform complete case analyses only. For most studies, only single items for suicidal ideation will be available. Single item measures are ordinal in nature. For ordinal outcomes, multiple imputation might not yield better results [50]. Additionally, with multiple imputation, we would introduce more noise (i.e., from the imputation error) to the noise already present in the data due to measurement error. Given that we will operationalize symptom deterioration and improvement based on single items, complete case analysis is the more conservative option.

## One-stage IPDMA

We will conduct a one-stage IPDMA, where all available IPD will be merged in a hierarchical model with participants nested in trials. This approach enables more sophisticated analyses compared to two-stage approaches, where estimates are calculated for each study separately in a first step [51] and then pooled in a second step. Importantly, the one-stage approach allows fine-grained moderator analyses [52]. We will model a random intercept in each of the models, reflecting the hierarchical structure of the data. The treatment effect may be modeled as fixed or random, depending on the homogeneity or heterogeneity of the treatment effect, respectively. The choice of model will be based on model comparisons via likelihood ratio tests. In case of heterogeneous treatment effects, we will allow random intercepts and slopes to correlate; the correlation will be estimated. All random effects in each model will be assumed to be multivariately normally distributed, with covariances between the random effects allowed and estimated. Treatment will be dummy-coded.

For the analyses of treatment response, we will conduct a multilevel logistic regression (response vs. no response). Single-item measures will be used for this analysis, including studies that provide a full measure of STB to increase comparability across trials. We will exclude all participants who did not report baseline STB. Response will be defined as STB reduction of at least one quartile in the item at post-intervention.

For the analyses of symptom deterioration, we will conduct a multilevel logistic regression (deterioration vs. no deterioration), analogous to the analysis of treatment response. Thus, we will use single-item measures from all available studies. Deterioration will be defined as at least one quartile deterioration in the respective item from baseline to post-intervention per participant. Participants who report the maximum score in STB at baseline will be excluded from the analysis because they cannot show deterioration.

For the corresponding logistic models, we are going to model the binary response $Y_{ij}$ (for person $j$ in study $i$; could be response or deterioriation) as

$$\ln\left(\frac{E(Y_{ij})}{1-E(Y_{ij})}\right) = \ln\left(\frac{p_{ij}}{1-p_{ij}}\right) = \alpha_i + \beta_{treat}x_{ij}. \tag{1}$$

It is $x_{ij} = 0$ for the control, and $x_{ij} = 1$ for the treatment group. We assume $\alpha_i \sim N\left(\alpha, \sigma_\alpha^2\right)$, where $\alpha$ is the fixed effect intercept and $\sigma_\alpha^2$ is the random effect variance for the intercept. If model comparisons indicate that treatment effects should be modeled heterogeneously, we will apply an extended version of (1) which includes a random treatment effect $\beta_{treat,i}$ rather than just the fixed effect $\beta_{treat}$ and assumes $(\alpha_i, \beta_{treat})' \sim \text{MVN}(\alpha, \beta_{treat})', \Sigma_1)$, with

$$\Sigma_1 = \begin{pmatrix} \sigma_\alpha^2 & \sigma_{\alpha,\beta_{treat}} \\ \sigma_{\alpha,\beta_{treat}} & \sigma_{\beta_{treat}}^2 \end{pmatrix}. \tag{2}$$

Furthermore, we will analyze continuous scores from those studies that provide full-scale STB measures. We will shift the scores to the same scale starting point and scale the post-intervention scores to the study-specific variance to ensure that scales are comparable across studies. We then employ multilevel regression models to predict the continuous post STB scores (for person $j$ in study $i$) as

$$y_{post,ij} = \alpha_i + \beta_{pre}y_{pre,ij} + \beta_{treat}x_{ij} + e_j. \tag{3}$$

It is $y_{post,ij}$ the post STB score, $y_{pre,ij}$ the baseline STB score (to control for it), and $x_{ij} = 0$ for the control, and $x_{ij} = 1$ for the treatment group. The intercept will again be modeled as random. The baseline effect will be modeled as either fixed ($\beta_{pre}$ as shown in (3)) or random (with $\beta_{pre,i}$ instead of $\beta_{pre}$ in (3)), relying on likelihood ratio tests to indicate which is more appropriate. In the same way, likelihood ratio tests will also guide the decision here whether the treatment effect should be modeled as fixed ($\beta_{treat}$ as shown in (3)) or random (with $\beta_{treat,i}$ instead of $\beta_{treat}$ in (3)). All random effects in each model (with model comparisons having indicated which to include of the afore described) will be assumed to be jointly multivariately normally distributed, with covariances between the random effects allowed and estimated. In the case of convergence problems, we will reduce the models in their random effects structure, prioritizing the inclusion of a random treatment effect if indicated. Furthermore, we assume $e_j \sim N(0, \sigma^2)$, that is, one homogenous error term across studies to simplify the model structure since this analysis will only be conducted with the subgroup of studies providing a continuous measure of STB.

Furthermore, if feasible, we will investigate the effects on suicidal behavior using logistic multilevel regression. For this outcome, we will combine both suicide attempts and deaths to a binary outcome (i.e., either at least one suicide attempt during the intervention period, or no suicide attempt). The logistic model will be fitted analogous to the models for response and deterioration.

The (generalized) linear mixed models for the one-stage models will be estimated using the lme4 package in R [53]. The lme4 estimates (generalized) linear mixed models using REML estimation for linear mixed models (for continuous score outcomes) and maximum likelihood for generalized linear mixed models (for binary outcomes, i.e., improvement and deterioration).

## Two-stage IPDMA

In addition, we will perform two-stage IPDMAs to examine the IPD separately in each study and then combine the estimates to calculate the pooled effect sizes. We will perform a separate two-stage IPDMA for all three indices of STB. As a result, we will compute Hedges' g for the severity of STB, as well as the logarithm of odds ratios for both treatment response and symptom deterioration. Based on the analysis of treatment response, we will calculate the number-needed-to-treat (NNT). This approach will enable us to compare our proposed IPDMA with previous and future meta-analyses. We will assess the degree of heterogeneity between studies using the $I^2$ statistic; we will use the restricted maximum-likelihood estimator for $\tau^2$. These analyses will be conducted using the R package metafor [54]. Any potential differences between the findings of the one- and two-stage approach will be discussed in the final manuscript in light of methodological differences [51].

## Moderators of STB severity

If feasible, we will conduct exploratory moderator analyses of whether the effects on STB may be influenced by the characteristics of the studies, interventions, or participants as suggested by prior research [2, 33, 55–66]. These analyses will be conducted in the one-stage IPDMA. We will perform moderator analyses using the following variables if feasible across the identified trials: (a) clinical variables at baseline and changes from baseline, including depressive symptoms, hopelessness, anxiety, alcohol use and psychiatric medication, (b) sociodemographic variables, including age, gender, level of education, relationship status, employment status, treatment history, and current or previous side treatments, (c) study-level variables including human support in the interventions, treatment duration, number of modules, and type of control condition.

The one-stage models for the respective measures will be extended through the inclusion of $\beta_{mod} z_{mod,ij}$ and the moderator-treatment interaction $\beta_{modXtreat} x_{ij} z_{mod,ij}$, with respective moderator $z_{mod}$. We will include one moderator at a time, fittings as many models as we have moderators, to avoid considerable reduction of eligible studies which include certain combinations of moderators all at once. Again, we are going to use model comparisons via likelihood ratio tests to determine where random effects are appropriate. In all models, all included random effects will be allowed to correlate.

## Sensitivity analyses

We will perform the following sensitivity analyses: First, we will exclude all trials that are overall rated as having a high risk of bias. Second, we will exclude trials with sample sizes below the median to investigate potential small-study effects. Third, for each single-item measure separately used multiple times across studies, we will conduct the single-item analyses (i.e., response and deterioration) separately to determine whether the findings are robust across different measures. This approach will allow us to identify potential differences across measures, which may arise from variations in the number of levels in the items. Fourth, the single-item analyses on symptom improvement and deterioration will be performed on all participants. Fifth, we will exclude trials in adolescents.

## Discussion

Depression remains a significant challenge to healthcare systems worldwide, and digital interventions are increasingly considered as a scalable treatment option, particularly when first-line

treatments are not available or desired. However, digital interventions for depression are not commonly recommended for people with STB.

This IPDMA aims to synthesize relevant data to provide information on the impact of digital interventions for depression on STB. A key strength of the IPDMA method in this context is its ability to capture data on STB from single items of depression scales that may not be reported in the original publications. The planned analyses on moderators of this effect will provide insights into potential beneficial or harmful effects in particular subgroups of patients, thereby contributing to the informed selection of digital interventions for depression.

This investigation has some limitations that should be acknowledged. Firstly, specific questionnaires on STB are expected to be available in only a few trials, meaning that the main analyses will focus on single items extracted from depression scales (e.g., item 9 of the PHQ-9). This reduces the degree of granulation, increases the error of the effect size estimation, and consequently reduces the content validity. Secondly, individuals who report elevated STB at baseline are often excluded from randomized clinical trials [27], which reduces the baseline symptom severity in the study population and therefore leaves less room for improvement compared to trials where STB are the primary outcome. Additionally, we will exclude people without STB at baseline from our response analysis, further reducing the total number of individuals in the planned analysis. Thirdly, the planned meta-analysis will combine data from scales created for adolescents and those created for adults, or scales that were applied in adolescents but may only be validated in adults. This may introduce bias to the analysis, which will be discussed in the final report. Fourthly, while suicidality is a critical aspect of depression, other psychopathologies, such as anxiety, substance abuse, and eating disorders, may also warrant investigation in future IPD projects. However, if this project proves successful, it may serve as a useful model for similar investigations in other psychopathologies in future IPD projects, contributing to a better understanding of the potential benefits and harms of digital interventions in mental health conditions. Fourthly, the database does not provide data on participants who have been withdrawn from studies due to suicidal risk or behavior. We will discuss this as a potential risk of bias in the final report.

Despite these limitations, this IPDMA is the first meta-analytic investigation to analyze the effects of digital cognitive behavioral therapy for depression on STB in the majority of available trials. The results of this study will offer crucial knowledge regarding the benefits and risks of providing digital interventions to individuals who experience suicidal thoughts in the context of depression.

## Supporting information

**S1 Checklist. PRISMA-P 2015 checklist.**
(PDF)

## Author Contributions

**Conceptualization:** Lasse Bosse Sander, Wouter van Ballegooijen, Pim Cuijpers, Tobias Teismann, Philipp Doebler, Matthias Domhardt, Harald Baumeister, Rebekka Büscher.

**Data curation:** Eirini Karyotaki, Wouter van Ballegooijen, Pim Cuijpers, Tobias Teismann, Philipp Doebler, Matthias Domhardt, Harald Baumeister, Rebekka Büscher.

**Formal analysis:** Wouter van Ballegooijen, Pim Cuijpers, Tobias Teismann, Philipp Doebler, Matthias Domhardt, Harald Baumeister, Rebekka Büscher.

**Funding acquisition:** Lasse Bosse Sander, Wouter van Ballegooijen, Pim Cuijpers, Tobias Teismann, Philipp Doebler, Matthias Domhardt, Harald Baumeister, Rebekka Büscher.

**Investigation:** Wouter van Ballegooijen, Pim Cuijpers, Tobias Teismann, Philipp Doebler, Matthias Domhardt, Harald Baumeister, Rebekka Büscher.

**Methodology:** Lasse Bosse Sander, Marie Beisemann, Eirini Karyotaki, Wouter van Ballegooijen, Pim Cuijpers, Tobias Teismann, Philipp Doebler, Matthias Domhardt, Harald Baumeister, Rebekka Büscher.

**Project administration:** Lasse Bosse Sander, Wouter van Ballegooijen, Pim Cuijpers, Tobias Teismann, Philipp Doebler, Matthias Domhardt, Harald Baumeister, Rebekka Büscher.

**Resources:** Wouter van Ballegooijen, Pim Cuijpers, Tobias Teismann, Philipp Doebler, Matthias Domhardt, Harald Baumeister, Rebekka Büscher.

**Software:** Wouter van Ballegooijen, Pim Cuijpers, Tobias Teismann, Philipp Doebler, Matthias Domhardt, Harald Baumeister, Rebekka Büscher.

**Supervision:** Lasse Bosse Sander, Marie Beisemann, Eirini Karyotaki, Wouter van Ballegooijen, Pim Cuijpers, Tobias Teismann, Philipp Doebler, Matthias Domhardt, Harald Baumeister, Rebekka Büscher.

**Validation:** Wouter van Ballegooijen, Pim Cuijpers, Tobias Teismann, Philipp Doebler, Matthias Domhardt, Harald Baumeister, Rebekka Büscher.

**Visualization:** Wouter van Ballegooijen, Pim Cuijpers, Tobias Teismann, Philipp Doebler, Matthias Domhardt, Harald Baumeister, Rebekka Büscher.

**Writing – original draft:** Lasse Bosse Sander, Wouter van Ballegooijen, Pim Cuijpers, Tobias Teismann, Philipp Doebler, Matthias Domhardt, Harald Baumeister, Rebekka Büscher.

**Writing – review & editing:** Lasse Bosse Sander, Marie Beisemann, Wouter van Ballegooijen, Pim Cuijpers, Tobias Teismann, Philipp Doebler, Matthias Domhardt, Harald Baumeister, Rebekka Büscher.

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
