## [Decision Letter · Decision Letter 0]

27 Mar 2023

PONE-D-23-05433Effects of digital depression interventions on suicidal thoughts and behavior: Protocol for a systematic review and meta-analysis of individual participant data

PLOS ONE

Dear Dr. Sander,

Thank you for submitting your manuscript to PLOS ONE. After careful consideration, we feel that it has merit but does not fully meet PLOS ONE’s publication criteria as it currently stands. Therefore, we invite you to submit a revised version of the manuscript that addresses the points raised during the review process.

ACADEMIC EDITOR:

Thank you and please take note of the suggestions. We find this paper interesting and provide good insight, overall. However, we offer some forms of suggestions and we look forward to hearing from you again.

A rebuttal letter that responds to each point raised by the academic editor and reviewer(s). You should upload this letter as a separate file labeled 'Response to Reviewers'.A marked-up copy of your manuscript that highlights changes made to the original version. You should upload this as a separate file labeled 'Revised Manuscript with Track Changes'.An unmarked version of your revised paper without tracked changes. You should upload this as a separate file labeled 'Manuscript'

We look forward to receiving your revised manuscript.

Kind regards,

Pei Boon Ooi, Ph.D.

Academic Editor

PLOS ONE

Journal Requirements:

2. We note that the original protocol file you uploaded contains a confidentiality notice indicating that the protocol may not be shared publicly or be published. Please note, however, that the PLOS Editorial Policy requires that the original protocol be published alongside your manuscript in the event of acceptance. Please note that should your paper be accepted, all content including the protocol will be published under the Creative Commons Attribution (CC BY) 4.0 license, which means that it will be freely available online, and any third party is permitted to access, download, copy, distribute, and use these materials in any way, even commercially, with proper attribution.

Therefore, we ask that you please seek permission from the study sponsor or body imposing the restriction on sharing this document to publish this protocol under CC BY 4.0 if your work is accepted. We kindly ask that you upload a formal statement signed by an institutional representative clarifying whether you will be able to comply with this policy. Additionally, please upload a clean copy of the protocol with the confidentiality notice (and any copyrighted institutional logos or signatures) removed."

"The study described in this protocol is funded by the German Research Foundation (project “Effects of internet interventions for depression on suicide ideation - a meta-analysis of individual participant data (IPDMA), grant No. SA 3767/2-1). The funders had no role in the study design, the decision to publish, or in the preparation of the manuscript. The views expressed are those of the author(s) and not necessarily those of the funder. The article processing charge was funded by the German Research Foundation (DFG) and the University of Freiburg in the funding program Open Access Publishing"

"LBS received funding for the study described in this protocol from the German Research Foundation (project “Effects of internet interventions for depression on suicide ideation - a meta-analysis of individual participant data (IPDMA), grant No. SA 3767/2-1).

http://www.dfg.de

The funders had no role in the study design, the decision to publish, or in the preparation of the manuscript. The views expressed are those of the author(s) and not necessarily those of the funder. The article processing charge was funded by the German Research Foundation (DFG) and the University of Freiburg in the funding program Open Access Publishing."

4. We note that you have stated that you will provide repository information for your data at acceptance. Should your manuscript be accepted for publication, we will hold it until you provide the relevant accession numbers or DOIs necessary to access your data. If you wish to make changes to your Data Availability statement, please describe these changes in your cover letter and we will update your Data Availability statement to reflect the information you provide

Additional Editor Comments (if provided):

Thank you and please take note of the suggestions.

Reviewers' comments:

Reviewer's Responses to Questions

**Comments to the Author**

1. Does the manuscript provide a valid rationale for the proposed study, with clearly identified and justified research questions?

Reviewer #1: Yes

Reviewer #2: Yes

2. Is the protocol technically sound and planned in a manner that will lead to a meaningful outcome and allow testing the stated hypotheses?

Reviewer #1: Yes

Reviewer #2: Yes

3. Is the methodology feasible and described in sufficient detail to allow the work to be replicable?

Reviewer #1: Yes

Reviewer #2: Yes

4. Have the authors described where all data underlying the findings will be made available when the study is complete?

Reviewer #1: No

Reviewer #2: Yes

5. Is the manuscript presented in an intelligible fashion and written in standard English?

Reviewer #1: Yes

Reviewer #2: No

6. Review Comments to the Author

You may also provide optional suggestions and comments to authors that they might find helpful in planning their study.

Reviewer #1: Summary

Thank you for the opportunity to review this protocol for a highly interesting and timely individual patient data meta-analysis. The authors have presented an excellent protocol which should be informative for readers considering the piece of work they will now go on to conduct. There are a number of issues that I think need addressing before it is publishable all of which should be relatively easily addressed.

Major Comments

1. If only digital CBT studies are going to be included in the dataset perhaps the title should be changed to reflect this type of treatment rather than broadly referring to “digital depression interventions”?

2. The “Population” subsection under “Eligibility Criteria” is not particularly detailed. Could the authors detail the age ranges they consider for definitions of adolescence and adulthood, and give some indication of whether or not they are seeking studies of samples with Major Depressive Disorder / a Major Depressive Episode or those with some symptoms of depression (e.g. above the “Minimal” range on the PHQ-9). In both cases (age and depression symptom severity) it could be argued that there are fundamental differences between the groups such that they might be best analysed separately. Conversely, it may be fine to include all patients together, but some rationale for this is needed.

3. One issue regarding the different age ranges is that a number of scales (e.g. Beck Scale for Suicidal Intent; PHQ-9) are only validated on adults. Combining outcomes from scales created for adolescents in particular with those created for adults may or may not be appropriate, regardless, doing so will introduce bias which needs to be considered and discussed.

4. The authors note that they will “conduct sensitivity analyses to check for the robustness of results”, could they give more details of the planned sensitivity analyses. One that they may consider might be to consider sub-group effects, e.g. analysing data for adolescents separately from adults, and also sub-grouping by the type of measure used to capture STB. For example, a number of the outcome measures used ask about suicidal ideation but not about any suicidal behaviour, others rate suicidal ideation and behaviour on the same item, and others still capture both ideation and behaviour in more broad scales. Further, some of the scales ask about symptoms in a narrow time-window (e.g. two weeks on PHQ-9) whereas others cover different time periods. So, if there are sufficient data to analyses the effects using the different ways of capturing STB or the different facets of STB that are captured in the different outcome measures, it would informative and helpful as a test of the robustness of the findings from the primary analyses.

5. The authors rightly note issues that might lead to a potential floor effect, however, there might also be a ceiling effect. I believe the authors have intended to do something about this (lines 261-264 on Page 12 but it is not entirely clear). It would help ensure findings are more robust if the authors are able to analyse data on individual participants that might have been withdrawn from studies due to suicidal risk/behaviour (following individual trial safety protocols) or on those that died by suicide (or had accidental death or open verdict recorded on their death certificate) during the study. If data on these individuals are not available (which would prevent the authors from analysing this in their one-stage models) but there are data at the study level on such withdrawals these might be included in two-stage models.

6. Can the authors comment on the number of studies that might be included in the systematic review but for which the authors have not / might not be able to access IPD, this is a common limitation in many IPD studies.

7. Can the authors give some rationale for their choice of potential moderators, many of these are associated with depression treatment outcomes, and all appear interesting but the rationale would help assure readers that this is not a ‘fishing exercise’ and is instead hypothesis driven.

8. The authors make no mention of missing data and how they will handle any. This is particularly important, especially as different studies will have different variables with which to impute missing data. As such, any imputation techniques are likely to be best applied at the single study level before aggregating the data into a single dataset, however this might have implications for the one-stage modelling.

9. There are no details given about the estimation methods1 and choice of specification for the residual variances,2 these can have important effects on the models produced and can lead to important differences in one-stage vs two-stage models. 3–5

References

1 Langan D, Higgins JPT, Jackson D, et al. A comparison of heterogeneity variance estimators in simulated random-effects meta-analyses. Res Synth Methods 2019; 10: 83–98.

2 Kovalchik SA, Cumberland WG. Using aggregate data to estimate the standard error of a treatment-covariate interaction in an individual patient data meta-analysis. Biometrical J 2012; 54: 370–84.

3 Debray TPA, Moons KGM, Abo-Zaid GMA, Koffijberg H, Riley RD. Individual Participant Data Meta-Analysis for a Binary Outcome: One-Stage or Two-Stage? PLoS One 2013; 8: e60650.

4 Riley RD, Legha A, Jackson D, et al. One‐stage individual participant data meta‐analysis models for continuous and binary outcomes: Comparison of treatment coding options and estimation methods. Stat Med 2020; 39: 2536–55.

5 Burke DL, Ensor J, Riley RD. Meta-analysis using individual participant data: one-stage and two-stage approaches, and why they may differ. Stat Med 2017; 36: 855–75.

Reviewer #2: Overall, I think this is a promising paper. The paper proposes protocols to let future research use IPD as a benchmarking tool to study STB. The paper also outlines the potential benefits and risks of delivering digital interventions for people who experience suicidal thoughts in the context of depression. However, I am quite confused about the selection criteria for the relevant sources to be included in the meta-analysis, and I think restructuring the contents may be helpful. Additionally, I am not sure why the researcher only focuses on depression, as other psychopathologies might be related to STB as well. Perhaps further clarification would be helpful to clear up any doubts. Also, I would suggest the manuscript undergo professional proofreading.

7. PLOS authors have the option to publish the peer review history of their article (what does this mean?). If published, this will include your full peer review and any attached files.

Reviewer #1: No

Reviewer #2: No

---

## [Author Response · Author response to Decision Letter 0]

19 Apr 2023

Dear anonymous reviewers of PLOS ONE,

Thank you for taking the time to review our manuscript and for providing such valuable feedback. We appreciate your insightful and stimulating comments, which have helped us to substantially and thoroughly revise the manuscript.

We have carefully addressed each of the points raised by the reviewers and believe that the manuscript has been significantly improved as a result. Attached, please find a detailed response to each of the reviewers' comments.

Once again, we sincerely thank you for your time and consideration.

Best regards,

Lasse B Sander

Point-by-point reply 

Reviewer #1

Summary

Thank you for the opportunity to review this protocol for a highly interesting and timely individual patient data meta-analysis. The authors have presented an excellent protocol which should be informative for readers considering the piece of work they will now go on to conduct. There are a number of issues that I think need addressing before it is publishable all of which should be relatively easily addressed.

Reply: We appreciate the reviewer for acknowledging the quality of our manuscript.

Major Comments

1. If only digital CBT studies are going to be included in the dataset perhaps the title should be changed to reflect this type of treatment rather than broadly referring to “digital depression interventions”?

Reply: Thank you for your comment. We have revised the title and made changes to the wording throughout the manuscript.

Revised title: Effects of digital cognitive behavioral therapy for depression on suicidal thoughts and behavior: Protocol for a systematic review and meta-analysis of individual participant data

2. The “Population” subsection under “Eligibility Criteria” is not particularly detailed. Could the authors detail the age ranges they consider for definitions of adolescence and adulthood, and give some indication of whether or not they are seeking studies of samples with Major Depressive Disorder / a Major Depressive Episode or those with some symptoms of depression (e.g. above the “Minimal” range on the PHQ-9). In both cases (age and depression symptom severity) it could be argued that there are fundamental differences between the groups such that they might be best analysed separately. Conversely, it may be fine to include all patients together, but some rationale for this is needed.

Reply: Thank you very much for raising this important point. We will consider studies in adults and adolescents from the age of 13 with symptoms of depression which may include individuals with a diagnosis of Major Depressive Disorder / a Major Depressive Episode. To give a better overview of our Eligibility Criteria, we added a textbox showing the PICOS criteria. 

Changes to the manuscript (added Textbox): 

Table 1. PICOS (participants, interventions, comparators, outcomes, study design) elements of the study inclusion criteria. 

Participants

 Adults or adolescents with depressive symptoms

Interventions

 Digital (web- or mobile-based) cognitive behavioral therapy interventions for depression with or without human support (guidance).

Comparators

• Usual care

• Attention control/psychological placebo

• Active control group

• No intervention

• Waitlist

Outcomes

 Depressive symptoms assessed by diagnostic interviews, self-reported and clinician -rated scales

 Suicidal thoughts and behaviors

- Self-report scales

- Diagnostic interviews

- Single items of depression scales (e.g. item 9 of the, PHQ-9) 

Study design

 Randomized controlled trials published in peer-reviewed journals.

We also agree on the importance on the differences between both mentioned groups (age and symptom severity). Therefore, we will conduct moderator analyses to investigate the potential moderating effect of these variables as stated in the section ’Moderators of STB severity’. Also referring to your comment #7, we revised the ’Moderators of STB severity’ section as follows: 

Revised section ’Moderators of STB severity’: If feasible, we will conduct exploratory moderator analyses of whether the effects on STB may be influenced by the characteristics of the studies, interventions, or participants as suggested by prior research [2,55–67]. These analyses will be conducted in the one-stage IPDMA. We will perform moderator analyses using the following variables if feasible across the identified trials: (a) clinical variables at baseline and changes from baseline, including depressive symptoms, hopelessness, anxiety, alcohol use and psychiatric medication, (b) sociodemographic variables, including age, gender, level of education, relationship status, employment status, treatment history, and current or previous side treatments, (c) study-level variables including human support in the interventions, treatment duration, number of modules, and type of control condition.

3. One issue regarding the different age ranges is that a number of scales (e.g. Beck Scale for Suicidal Intent; PHQ-9) are only validated on adults. Combining outcomes from scales created for adolescents in particular with those created for adults may or may not be appropriate, regardless, doing so will introduce bias which needs to be considered and discussed.

Reply: We appreciate the reviewer for this insightful comment, and we have added this important point to the discussion section as a potential limitation of our planned analysis. In addition, we added a sensitivity analysis in which studied in adolescents will be excluded. 

Changes to the manuscript (Sensitivity analyses section): Fifth, we will exclude trials in adolescents.

Changes to the manuscript (discussion section): Thirdly, the planned meta-analysis will combine data from scales created for adolescents and those created for adults, or scales that were applied in adolescents but may only be validated in adults. This may introduce bias to the analysis, which will be discussed in the final report. 

4. The authors note that they will “conduct sensitivity analyses to check for the robustness of results”, could they give more details of the planned sensitivity analyses. One that they may consider might be to consider sub-group effects, e.g. analysing data for adolescents separately from adults, and also sub-grouping by the type of measure used to capture STB. For example, a number of the outcome measures used ask about suicidal ideation but not about any suicidal behaviour, others rate suicidal ideation and behaviour on the same item, and others still capture both ideation and behaviour in more broad scales. Further, some of the scales ask about symptoms in a narrow time-window (e.g. two weeks on PHQ-9) whereas others cover different time periods. So, if there are sufficient data to analyses the effects using the different ways of capturing STB or the different facets of STB that are captured in the different outcome measures, it would informative and helpful as a test of the robustness of the findings from the primary analyses.

Reply: Thank you for the comment. We agree with the reviewer and have elaborated on the sensitivity analyses in the manuscript (paragraph: ‘Sensitivity analyses’). We have deleted the sentence on the sensitivity analysis from the ‘statistical analyses’ section to make the methods section more concise. We will investigate the potential influence of age in the moderator analyses as stated under ‘moderators of STB severity ‘.

Revised section ‘Sensitivity analyses’: We will perform the following sensitivity analyses: First, we will exclude all trials that are overall rated as having a high risk of bias. Second, we will exclude trials with sample sizes below the median to investigate potential small-study effects. Third, for each single-item measure separately used multiple times across studies, we will conduct the single-item analyses (i.e., response and deterioration) separately to determine whether the findings are robust across different measures. This approach will allow us to identify potential differences across measures, which may arise from variations in the number of levels in the items. Fourth, the single-item analyses on symptom improvement and deterioration will be performed on all participants. Fifth, we will exclude trials in adolescents.

5. The authors rightly note issues that might lead to a potential floor effect, however, there might also be a ceiling effect. I believe the authors have intended to do something about this (lines 261-264 on Page 12 but it is not entirely clear). It would help ensure findings are more robust if the authors are able to analyse data on individual participants that might have been withdrawn from studies due to suicidal risk/behaviour (following individual trial safety protocols) or on those that died by suicide (or had accidental death or open verdict recorded on their death certificate) during the study. If data on these individuals are not available (which would prevent the authors from analysing this in their one-stage models) but there are data at the study level on such withdrawals these might be included in two-stage models.

Reply: Thank you for this important point. Unfortunately, the IPD dataset does not include this data, but we have added this to the limitations section to be aware of this potential bias. 

Changes to the manuscript (discussion section): Fourthly, the database does not provide data on participants who have been withdrawn from studies due to suicidal risk or behavior. We will discuss this as a potential risk of bias in the final report.

6. Can the authors comment on the number of studies that might be included in the systematic review but for which the authors have not / might not be able to access IPD, this is a common limitation in many IPD studies.

Reply: We fully agree on this important possible limitation. The most recent publication from the database (Karyotaki 2021) included individual patient data (IPD) from 39 out of 42 eligible randomized controlled trials (RCTs). The last update has led to IPD from 62 studies. We added this information to the introduction section. However, the database is currently undergoing a major update, and as of now, any estimate we provide on the ratio of eligible studies vs. studies providing IPD would be purely speculative. However, we will comment on this very important point in the final study report as described in the last paragraph of the ‘risk of bias section’ (availability bias).

Revised ‘Risk of bias’ section: Moreover, we will assess availability bias by examining whether the studies included in this IPDMA with available IPD and STB measures are representative of all relevant trials. First, we will conduct a random-effects meta-analysis of aggregated data with depressive symptoms as the outcome using the classic inverse-variance approach, performing a subgroup analysis to compare studies included in this IPDMA with all other relevant trials that did not provide IPD with STB measure. The between-groups effect will be calculated using change scores from baseline. Second, we will display study characteristics for included trials and other relevant trials, such as year of publication, treatment format, number of sessions, age, gender, recruitment strategy, control condition, to identify any group differences.

Introduction section referring to this comment: The proposed IPDMA will leverage an existing IPD pool of currently 62 studies (14,088 participants) on i-CBT interventions for depression [20,38,39]. Currently, this dataset is undergoing a major update, and the number of included studies and participants is expected to alter.

Reference:

Karyotaki E, Efthimiou O, Miguel C, et al. Internet-Based Cognitive Behavioral Therapy for Depression: A Systematic Review and Individual Patient Data Network Meta-analysis. JAMA Psychiatry. 2021;78(4):361–371. doi:10.1001/jamapsychiatry.2020.4364

7. Can the authors give some rationale for their choice of potential moderators, many of these are associated with depression treatment outcomes, and all appear interesting, but the rationale would help assure readers that this is not a ‘fishing exercise’ and is instead hypothesis driven.

Reply: We appreciate the reviewer for bringing this issue into discussion. Accordingly, we have marked the moderator analyses as exploratory and will make interpretations with caution (e.g. mark them as preliminary) to provide hints for future confirmatory studies. Furthermore, we added references for possible moderators, which were identified to be associated with the outcome in prior studies in the context of STB, depression, and digital interventions. 

Changes to the manuscript (methods section): If feasible, we will conduct exploratory moderator analyses of whether the effects on STB may be influenced by the characteristics of the studies, interventions, or participants as suggested by prior research [2,55–67].

Reference added to the manuscript: 

Karyotaki E, Efthimiou O, Miguel C, Bermpohl FM genannt, Furukawa TA, Cuijpers P, et al. Internet-based cognitive behavioral therapy for depression: A systematic review and individual patient data network meta-analysis. JAMA Psychiatry. 2021;78: 361–371. doi:10.1001/jamapsychiatry.2020.4364

Domhardt M, Steubl L, Boettcher J, Buntrock C, Karyotaki E, Ebert D, et al. Mediators and mechanisms of change in internet- and mobile-based interventions for depression: A systematic review. Clin Psychol Rev. 2021;83: 101953. doi:10.1016/j.cpr.2020.101953

Furukawa TA, Suganuma A, Ostinelli EG, Andersson G, Beevers CG, Shumake J, et al. Dismantling, optimising, and personalising internet cognitive behavioural therapy for depression: A systematic review and component network meta-analysis using individual participant data. Lancet Psychiatry. 2021;8: 500–511. doi:10.1016/S2215-0366(21)00077-8

Handley TE, Kay-Lambkin FJ, Baker AL, Lewin TJ, Kelly BJ, Inder KJ, et al. Incidental treatment effects of CBT on suicidal ideation and hopelessness. J Affect Disord. 2013;151: 275–283. doi:10.1016/j.jad.2013.06.005

Witt KG, Hetrick SE, Rajaram G, Hazell P, Taylor Salisbury TL, Townsend E, et al. Psychosocial interventions for self-harm in adults. Cochrane Database of Systematic Reviews. 2021;2021. doi:10.1002/14651858.CD013668.pub2

van Heeringen K, Mann JJ. The neurobiology of suicide. Lancet Psychiatry. 2014;1: 63–72. doi:10.1016/S2215-0366(14)70220-2

Ferrari AJ, Norman RE, Freedman G, Baxter AJ, Pirkis JE, Harris MG, et al. The burden attributable to mental and substance use disorders as risk factors for suicide: findings from the Global Burden of Disease Study 2010. PLoS One. 2014;9: e91936. doi:10.1371/journal.pone.0091936

Büscher R, Beisemann M, Doebler P, Micklitz HM, Kerkhof A, Cuijpers P, et al. Digital cognitive–behavioural therapy to reduce suicidal ideation and behaviours: A systematic review and meta-analysis of individual participant data. Evidence Based Mental Health. 2022;25: e8–e17. doi:10.1136/ebmental-2022-300540

Sobanski T, Josfeld S, Peikert G, Wagner G. Psychotherapeutic interventions for the prevention of suicide re-attempts: a systematic review. Psychol Med. 2021;51: 2525–2540. doi:10.1017/S0033291721003081

Franklin JC, Ribeiro JD, Fox KR, Bentley KH, Kleiman EM, Huang X, et al. Risk factors for suicidal thoughts and behaviors: A meta-analysis of 50 years of research. Psychol Bull. 2017;143: 187–232. doi:10.1037/bul0000084

Ballard ED, Ionescu DF, Vande Voort JL, Niciu MJ, Richards EM, Luckenbaugh DA, et al. Improvement in suicidal ideation after ketamine infusion: Relationship to reductions in depression and anxiety. J Psychiatr Res. 2014;58: 161–166. doi:10.1016/j.jpsychires.2014.07.027

Cougle JR, Keough ME, Riccardi CJ, Sachs-Ericsson N. Anxiety disorders and suicidality in the National Comorbidity Survey-Replication. J Psychiatr Res. 2009;43: 825–829. doi:10.1016/j.jpsychires.2008.12.004

Batterham PJ, Spijker BAJ, Mackinnon AJ, Calear AL, Wong Q, Christensen H. Consistency of trajectories of suicidal ideation and depression symptoms: Evidence from a randomized controlled trial. Depress Anxiety. 2019;36: 321–329. doi:10.1002/da.22863

8. The authors make no mention of missing data and how they will handle any. This is particularly important, especially as different studies will have different variables with which to impute missing data. As such, any imputation techniques are likely to be best applied at the single study level before aggregating the data into a single dataset, however this might have implications for the one-stage modelling.

Reply: This is a very important point and we discussed this intensively in the authors group. Finally, we decided to not impute missing data because for most studies, only single items for suicidal ideation will be available. Especially the single items will be affected by measuring error, which might get worse when using multiple imputation. In ordinal outcomes (i.e., the single item data), multiple imputation does not necessarily yield better results (ref Chen et al. 2005). This is especially relevant as we will operationalize symptom deterioration and improvement as a change in score by at least one quartile from baseline, which will be only one point difference on a single item in some cases. Thus, complete case analysis is the more conservative analysis.

Changes to the manuscript (Statistical analyses): With regard to missing data handling, we will only perform complete case analyses. For most studies, only single items for suicidal ideation will be available. Single item measures are ordinal in nature. For ordinal outcomes, multiple imputation might not yield better results [50]. Additionally, with multiple imputation, we would introduce more noise (i.e., from the imputation error) to the noise already present in the data due to measurement error. Given that we will operationalize symptom deterioration and improvement based on single items, complete case analysis is the more conservative option.

Reference added to the manuscript: Chen, Ling; Toma-Drane, Marian; Valois, Robert F.; and Drane, J. Wanzer (2005) "Multiple Imputation For Missing Ordinal Data," Journal of Modern Applied Statistical Methods: Vol. 4 : Iss. 1 , Article 26. DOI: 10.22237/jmasm/1114907160 Available at: http://digitalcommons.wayne.edu/jmasm/vol4/iss1/26

9. There are no details given about the estimation methods1 and choice of specification for the residual variances,2 these can have important effects on the models produced and can lead to important differences in one-stage vs two-stage models. 3–5

1 Langan D, Higgins JPT, Jackson D, et al. A comparison of heterogeneity variance estimators in simulated random-effects meta-analyses. Res Synth Methods 2019; 10: 83–98.

2 Kovalchik SA, Cumberland WG. Using aggregate data to estimate the standard error of a treatment-covariate interaction in an individual patient data meta-analysis. Biometrical J 2012; 54: 370–84.

3 Debray TPA, Moons KGM, Abo-Zaid GMA, Koffijberg H, Riley RD. Individual Participant Data Meta-Analysis for a Binary Outcome: One-Stage or Two-Stage? PLoS One 2013; 8: e60650.

4 Riley RD, Legha A, Jackson D, et al. One‐stage individual participant data meta‐analysis models for continuous and binary outcomes: Comparison of treatment coding options and estimation methods. Stat Med 2020; 39: 2536–55.

5 Burke DL, Ensor J, Riley RD. Meta-analysis using individual participant data: one-stage and two-stage approaches, and why they may differ. Stat Med 2017; 36: 855–75.

Reply: Thank you for bringing this inaccuracy to our attention. The (generalized) linear mixed models for the one-stage models will be estimated using the lme4 package in R (Douglas Bates, Martin Maechler, Ben Bolker, Steve Walker (2015). Fitting Linear Mixed-Effects Models Using lme4. Journal of Statistical Software, 67(1), 1-48. doi:10.18637/jss.v067.i01.).

In lme4, the lmer function estimates (generalized) linear mixed models using (by default) restricted maximum likelihood (REML) estimation for linear mixed models (for continuous score outcomes) and the glmer function uses maximum likelihood estimation for generalized linear mixed models (for binary outcomes, i.e., improvement and deterioration), please see Bates et al. and lme4 documentation (https://cran.r-project.org/web/packages/lme4/lme4.pdf) for details on the estimation procedure. 

Treatment will be dummy-coded when entered into these models. The hierarchical structure of the data will be accounted for using a random intercept. We opt for a random intercept rather than one (fixed) intercept per study as to not endanger the estimator’s consistency – adding a fixed intercept for each study implies that the number of model parameters will grow with the number of observations, or rather, studies (so called Neyman-Scott problem). For any model parameter, we therefore model study-specificity by using a random effect. 

As we unfortunately cannot expect large numbers of depression intervention studies to contain entire STB measures (i.e., continuous scores), we expect the focus of our analysis to be on studies with single-item measures for which we plan to record treatment response and deterioration from pre to post (described in manuscript). These will be modeled using logistic multilevel regression models (which are analogously also going to be used – if feasible – for the analysis of suicidal behavior). 

For the logistic models (for treatment response, deterioration, and if feasible suicidal behavior), we are going to model the 

binary response y_ij (for person j in study i) as 

ln⁡((E(Y_ij))/(1-E(Y_ij)))= ln⁡〖(p_ij/(1-p_ij ))= α_i+ β_treat x_ij 〗 (1)

It is x_ij=0 for the control, and x_ij=1 for the treatment group. As explained, we assume α_(i ) ~ N(α,σ_α^2 ), where α is the fixed effect intercept and σ_α^2 is the random effect variance for the intercept. We rely on a likelihood ratio test (note that for these model comparisons, models need to be re-fitted using ML estimation, which lme4 automatically takes care of during model comparisons) to assess whether we further model the treatment effect β_treat as random or fixed. Note that this model comparison answers the question whether treatments effect should be modeled homogenously (as a fixed effect β_treat) or heterogeneously (as a random effect β_(treat,i)). An extended version of (1) which includes a random treatment effect β_(treat,i) rather than just the fixed effect β_treat assumes (α_(i ),β_(treat,i))' ~ MVN((α,β_treat)',Σ_1), with

Σ_1=(■(σ_α^2&σ_(α,β_treat )@σ_(α,β_treat )&σ_(β_treat)^2 )). (2)

As explained, we also plan to analyze continuous scores from those studies which included full-scale SBT measures, with these models being a secondary focus of the planned analyses. We will shift the scores to the same scale starting point and scale the post-intervention scores to the study-specific variance to ensure that scales are comparable across studies. We then employ multilevel regression models to predict the continuous post STB scores (for person j in study i) as

y_(post,ij)= α_(i )+ β_pre y_(pre,ij)+ β_treat x_ij+ e_j. (3)

It is y_(post,ij) the post STB score, y_(pre,ij) the baseline STB score, and x_ij=0 for the control, and x_ij=1 for the treatment group. Again, we assume α_(i ) ~ N(α,σ_α^2 ), where α is the fixed effect intercept and σ_α^2 is the random effect variance for the intercept. We control for the baseline (pre) STB scores y_(pre,ij). Using likelihood ratio tests, we determine whether the baseline effect can be modeled as fixed as in Equation (3) or whether it should instead by modeled as random, by swapping β_pre for β_(pre,i) in Equation (3) and assuming α_(i ) and β_(pre,i) jointly (α_(i ),β_(pre,i))' ~ MVN((α,β_pre)',Σ_2), where β_pre denotes the fixed baseline effect and with

Σ_2=(■(σ_α^2&σ_(α,β_pre )@σ_(α,β_pre )&σ_(β_pre)^2 )), (4)

where σ_(β_pre)^2 is the random effects variance for β_(pre,i) and σ_(α,β_pre ) is the covariance between α_(i ) and β_(pre,i). In the same way, we rely on likelihood ratio tests to assess whether we further model the treatment effect β_treat as random or fixed. Note that this model comparison answers the question whether treatments effect should be modeled homogenously (as a fixed effect β_treat) or heterogeneously (as a random effect β_(treat,i)). Depending on whether we modeled β_pre as fixed (Σ_3) or random (Σ_4), we assume (α_(i ),β_(treat,i))' ~ MVN((α,β_treat)',Σ_3) or (α_(i ),β_(pre,i),β_(treat,i))' ~ MVN((α,〖 β〗_pre,β_treat)',Σ_4), with 

Σ_3=(■(σ_α^2&σ_(α,β_treat )@σ_(α,β_treat )&σ_(β_treat)^2 )) or Σ_4= (■(σ_α^2&σ_(α,β_pre )&σ_(α,β_treat )@σ_(α,β_pre )&σ_(β_pre)^2&σ_(β_pre,β_treat )@σ_(α,β_treat )&σ_(β_pre,β_treat )&σ_(β_treat)^2 )). (5)

In Equation (1), we assume e_j ~ N(0,σ^2), that is, one common (homogenous) error term across studies. This assumption was made to simplify the model structure for the continuous score models which are likely only going to contain a handful of studies and will not be the focus of our analyses. 

Note that these continuous-score models can become quite complex in their random effects structure if for every one of these decisions, we are going to choose to extend the random effects structure based on the model comparisons. This could be especially challenging if we only end up with a small number of studies which included continuous scores. If this leads to convergence issues, we are going to reduce the models in their random effects structure, prioritizing the inclusion of a random treatment effect if indicated so that we can adequately model heterogeneity in treatment effects.

For the moderator analyses, the chosen models for the respective measures will be extended through the inclusion of β_mod z_(mod,ij) and the moderator-treatment interaction β_modXtreat x_ij z_(mod,ij), with respective moderator z_mod. We will include one moderator at a time, fittings as many models as we have moderators, to avoid considerable reduction of eligible studies which include certain combinations of moderators all at once. Again, we are going to use model comparisons via likelihood ratio tests to determine where random effects are appropriate. In all models, all included random effects will be allowed to correlate.

For the two-stage and conventional meta-analysis, we are going to use traditional methods of meta-analysis on previously aggregated study-wise data. We are going to use the R package metafor (CITE: Viechtbauer, W. (2010). Conducting meta-analyses in R with the metafor package. Journal of Statistical Software, 36(3), 1-48. https://doi.org/10.18637/jss.v036.i03). The first stage will involve computing Hedges’ g for the severity of STB, as well as the logarithm of odds ratios for both treatment response and symptom deterioration for each study. The second step consists of the across-study aggregation, using random-effects inverse-variance estimation approach will be taken. Between-study heterogeneity will be estimated using the I2 statistic; we will use the restricted maximum-likelihood estimator (REML) for τ2. This analysis will be treated as a sensitivity analysis and can include even studies for which primary data was not available.

We report the outcome for both one- and two-stage approaches. We ideally expect qualitatively comparable results. Any potential differences between the two approaches are going to be discussed in the final manuscript in light of methodological differences between the approaches (Burke DL, Ensor J, Riley RD. Meta-analysis using individual participant data: one-stage and two-stage approaches, and why they may differ. Stat Med 2017; 36: 855–75).

Changes to the manuscript (Risk of bias section): Moreover, we will assess availability bias by examining whether the studies included in this IPDMA with available IPD and STB measures are representative of all relevant depression trials. First, we will conduct a random-effects meta-analysis of aggregated data with depressive symptoms as the outcome using the classic inverse-variance approach, performing a subgroup analysis to compare studies included in this IPDMA with all other relevant trials that did not provide IPD with STB measure. The between-groups effect will be calculated using change scores from baseline.

Changes to the manuscript (One-stage IPDMA section): The choice of model will be based on model comparisons via likelihood ratio tests. In case of heterogeneous treatment effects, we will allow random intercepts and slopes to correlate; the correlation will be estimated. All random effects in each model will be assumed to be multivariately normally distributed, with covariances between the random effects allowed and estimated. Treatment will be dummy-coded.

[…]

For the corresponding logistic models, we are going to model the binary response Y_ij (for person j in study i; could be response or deterioriation) as 

ln⁡((E(Y_ij))/(1-E(Y_ij)))= ln⁡〖(p_ij/(1-p_ij ))= α_i+ β_treat x_ij 〗. (1)

It is x_ij=0 for the control, and x_ij=1 for the treatment group. We assume α_(i ) ~ N(α,σ_α^2 ), where α is the fixed effect intercept and σ_α^2 is the random effect variance for the intercept. If model comparisons indicate that treatment effects should be modeled heterogeneously, we will apply an extended version of (1) which includes a random treatment effect β_(treat,i) rather than just the fixed effect β_treat and assumes (α_(i ),β_(treat,i))' ~ MVN((α,β_treat)',Σ_1), with

Σ_1=(■(σ_α^2&σ_(α,β_treat )@σ_(α,β_treat )&σ_(β_treat)^2 )). (2)

Furthermore, we will analyze continuous scores from those studies that provide full-scale SBT measures. We will shift the scores to the same scale starting point and scale the post-intervention scores to the study-specific variance to ensure that scales are comparable across studies. We then employ multilevel regression models to predict the continuous post STB scores (for person j in study i) as

y_(post,ij)= α_(i )+ β_pre y_(pre,ij)+ β_treat x_ij+ e_j. (3)

It is y_(post,ij) the post STB score, y_(pre,ij) the baseline STB score (to control for it)), and x_ij=0 for the control, and x_ij=1 for the treatment group. The intercept will again be modeled as random. The baseline effect will be modeled as either fixed (β_pre as shown in (3)) or random (with β_(pre,i) instead of β_pre in (3)), relying on likelihood ratio tests to indicate which is more appropriate. In the same way, likelihood ratio tests will also guide the decision here whether the treatment effect should be modeled as fixed (β_treat as shown in (3)) or random (with β_(treat,i) instead of β_treat in (3)). All random effects in each model (with model comparisons having indicated which to include of the afore described) will be assumed to be jointly multivariately normally distributed, with covariances between the random effects allowed and estimated. In the case of convergence problems, we will reduce the models in their random effects structure, prioritizing the inclusion of a random treatment effect if indicated. Furthermore, we assume e_j ~ N(0,σ^2), that is, one homogenous error term across studies to simplify the model structure since this analysis will only be conducted with the subgroup of studies providing a continuous measure of STB.

[…]

The logistic model will be fitted analogous to the models for response and deterioration. 

The (generalized) linear mixed models for the one-stage models will be estimated using the lme4 package in R [53]. The lme4 estimates (generalized) linear mixed models using REML estimation for linear mixed models (for continuous score outcomes) and maximum likelihood for generalized linear mixed models (for binary outcomes, i.e., improvement and deterioration).

Changes to the manuscript (Two-stage IPDMA): We will assess the degree of heterogeneity between studies using the I2 statistic; we will use the restricted maximum-likelihood estimator for τ2. These analyses will be conducted using the R package metafor [54]. Any potential differences between the findings of the one- and two-stage approach will be discussed in the final manuscript in light of methodological differences.

Changes to the manuscript (Moderators of STB severity): The one-stage models for the respective measures will be extended through the inclusion of β_mod z_(mod,ij) and the moderator-treatment interaction β_modXtreat x_ij z_(mod,ij), with respective moderator z_mod. We will include one moderator at a time, fittings as many models as we have moderators, to avoid considerable reduction of eligible studies which include certain combinations of moderators all at once. Again, we are going to use model comparisons via likelihood ratio tests to determine where random effects are appropriate. In all models, all included random effects will be allowed to correlate.

Reference added to the manuscript: 

Bates D, Mächler M, Bolker B, Walker S. Fitting Linear Mixed-Effects Models using lme4. 2014. Available: https://arxiv.org/abs/1406.5823

Viechtbauer W. Conducting Meta-Analyses in R with the metafor Package. J Stat Softw. 2010;36: 1–48. doi:10.18637/jss.v036.i03

Reviewer #2

Overall, I think this is a promising paper. The paper proposes protocols to let future research use IPD as a benchmarking tool to study STB. The paper also outlines the potential benefits and risks of delivering digital interventions for people who experience suicidal thoughts in the context of depression. 

Reply: We thank the reviewer for the acknowledgement of our protocol manuscript.

 However, I am quite confused about the selection criteria for the relevant sources to be included in the meta-analysis, and 

I think restructuring the contents may be helpful. 

Reply: Thank you for this critical comment. We have included a textbox listing the PICOS inclusion criteria to enhance clarity and improve the reader's understanding.

 Additionally, I am not sure why the researcher only focuses on depression, as other psychopathologies might be related to STB as well. Perhaps further clarification would be helpful to clear up any doubts. 

Reply: Thank you for this important comment. As outlined in the introduction section, depression is one of the leading challenges of healthcare settings worldwide. Suicidality is a key symptom of depression in the ICD and DSM diagnostic manuals. Additionally, the issue of patient safety in digital interventions is currently a topic of considerable discussion, particularly regarding whether this form of intervention is suitable for people with suicidal thoughts. As presented in both the introduction and discussion, our study aims to contribute to the question of whether digital interventions are appropriate for people with depression and suicidal ideation.

While we acknowledge that other psychopathologies may also be relevant in the context of STB, another main reason why we focus on depression is due to the existence of an IPD database for depression. Establishing such a database requires several years of work and a high degree of networking and is thus out of scope of our investigation. Nevertheless, we hope that our work can serve as a foundation for future research involving other psychopathologies.

According to the reviewer’s comment, we have added a section to the limitations to emphasize the relevance of other conditions in the context of suicidality. 

Changes to the manuscript (discussion section): Fourthly, while suicidality is a critical aspect of depression, other psychopathologies, such as anxiety, substance abuse, and eating disorders, may also warrant investigation in future IPD projects. However, if this project proves successful, it may serve as a useful model for similar investigations in other psychopathologies in future IPD projects, contributing to a better understanding of the potential benefits and harms of digital interventions in mental health conditions.

 Also, I would suggest the manuscript undergo professional proofreading.

Reply: We appreciate the reviewer for suggesting additional language proofreading, and we have performed the necessary revisions.

---

## [Editor Report · Decision Letter 1]

27 Apr 2023

Effects of digital cognitive behavioral therapy for depression on suicidal thoughts and behavior: Protocol for a systematic review and meta-analysis of individual participant data

PONE-D-23-05433R1

Dear Dr. Sander,

We’re pleased to inform you that your manuscript has been judged scientifically suitable for publication and will be formally accepted for publication once it meets all outstanding technical requirements.

Kind regards,

Pei Boon Ooi, Ph.D.

Academic Editor

PLOS ONE
---

## [Editor Report · Acceptance letter]

31 May 2023

PONE-D-23-05433R1 

Effects of digital cognitive behavioral therapy for depression on suicidal thoughts and behavior: Protocol for a systematic review and meta-analysis of individual participant data 

Dear Dr. Sander:

I'm pleased to inform you that your manuscript has been deemed suitable for publication in PLOS ONE. Congratulations! Your manuscript is now with our production department. 

Kind regards, 

on behalf of

Dr. Pei Boon Ooi 

Academic Editor

PLOS ONE